# Cardiac and Vascular α_1_-Adrenoceptors in Congestive Heart Failure: A Systematic Review

**DOI:** 10.3390/cells9112412

**Published:** 2020-11-04

**Authors:** Gizem Kaykı-Mutlu, Olga Papazisi, Meindert Palmen, A. H. Jan Danser, Martin C. Michel, Ebru Arioglu-Inan

**Affiliations:** 1Department of Pharmacology, Faculty of Pharmacy, Ankara University, 06560 Ankara, Turkey; gkayki@ankara.edu.tr (G.K.-M.); ebru.arioglu@ankara.edu.tr (E.A.-I.); 2Department of Cardiothoracic Surgery, Leiden University Medical Center, 2300 RC Leiden, The Netherlands; o.papazisi@lumc.nl (O.P.); M.Palmen@lumc.nl (M.P.); 3Department of Internal Medicine, Division of Pharmacology, Erasmus Medical Center, 3000 CA Rotterdam, The Netherlands; a.danser@erasmusmc.nl; 4Department of Pharmacology, Johannes Gutenberg University, 55131 Mainz, Germany

**Keywords:** α_1_-adrenoceptors, cardiomyocyte, blood vessel, heart failure

## Abstract

As heart failure (HF) is a devastating health problem worldwide, a better understanding and the development of more effective therapeutic approaches are required. HF is characterized by sympathetic system activation which stimulates α- and β-adrenoceptors (ARs). The exposure of the cardiovascular system to the increased locally released and circulating levels of catecholamines leads to a well-described downregulation and desensitization of β-ARs. However, information on the role of α-AR is limited. We have performed a systematic literature review examining the role of both cardiac and vascular α_1_-ARs in HF using 5 databases for our search. All three α_1_-AR subtypes (α_1A_, α_1B_ and α_1D_) are expressed in human and animal hearts and blood vessels in a tissue-dependent manner. We summarize the changes observed in HF regarding the density, signaling and responses of α_1_-ARs. Conflicting findings arise from different studies concerning the influence that HF has on α_1_-AR expression and function; in contrast to β-ARs there is no consistent evidence for down-regulation or desensitization of cardiac or vascular α_1_-ARs. Whether α_1_-ARs are a therapeutic target in HF remains a matter of debate.

## 1. Introduction

The sympathetic nervous system primarily regulates cardiac function via β-adrenoceptors (β-ARs) [1,2], whereas α_1_-adrenoceptors (α_1_-ARs) play an important role in the regulation of vascular tone [3]. Yet, α_1_-ARs are also expressed in various cell types of the heart and can contribute to the regulation of cardiac contraction [4,5], rhythm [6,7] and hypertrophy [8]. However, the role of cardiac α_1_-ARs appears to be species-dependent with a considerable contribution in rodents but a much smaller (if any) contribution in the healthy human heart [1]. On the other hand, cardiac expression of other subtypes such as β_3_-ARs also appears to be lower in the human heart than in that of other species, but such receptors become up-regulated in congestive heart failure (CHF) [9].

The sympatho-adrenal system is activated in acute and chronic heart failure (HF). Accordingly, plasma levels of noradrenaline and its co-transmitters are increased in CHF [10,11]. This appears to correlate with a higher risk of mortality [12]. In line with this, patients with reduced left ventricular function appear to be more prone to post-operative vasoplegia that additionally increases the mortality and morbidity risk [13,14]. The regulation of β-ARs in CHF has been reviewed extensively and appears distinct for the three β-AR subtypes [1].

α_1_-ARs are also a subfamily with three members: α_1A_, α_1B_ and α_1D_ [15]. Data outside the cardiovascular system suggest that they can undergo differential regulation upon extended agonist exposure [16]. Nevertheless, until now, the α_1_-AR regulation in HF has not been fully elucidated and results might vary a lot between different studies. Against this background we discuss the expression and function of α_1_-ARs and their subtypes in the heart and vasculature of patients with CHF and animal models thereof based upon a systematic review of the literature.

## 2. Materials and Methods

A broad literature search was performed in 5 databases (PubMed/MEDLINE, Embase, Web of Science, Cochrane Library, Emcare). The search was performed using the MeSH terms “Heart failure” and “Receptors, Adrenergic alpha 1” and by combining each term with different keyword variations and free text words. No language limits were applied. The final search was performed on the 15^th^ of April 2020. Details of the search strategy are provided in the Appendix A.

Animal and human studies were included that investigated changes of cardiac and/or vascular α_1_-AR expression or function in CHF. There were no limitations regarding the study design or the etiology/model of CHF. Two independent reviewers (GKM and OP) assessed all articles for eligibility that the search strategy had yielded. The first screening was performed by assessing only the titles and abstracts of the articles, while the final decision for inclusion was based upon reading the full text of the selected articles. The reference lists of all relevant articles and reviews were manually checked for inclusion of additional eligible studies that were not identified by the primary search strategy. Any disagreements that emerged during the study selection process were resolved by discussion. A Preferred Reporting Items for Systematic Reviews and Meta-Analyses (PRISMA, www.prisma-statement.org) flowchart is shown as Figure 1.

The data from the eligible studies were retrieved and recorded in predefined Google Spreadsheet forms that offer the possibility of auditing. Data collection included study characteristics and study design information and changes that were identified regarding the expression and function of α_1_-ARs in cells, tissue and in vivo.

## 3. Considerations Related to Data Interpretation

Most investigators have reported mRNA levels after normalization to a reference gene, often glyceraldehyde-3-phosphate dehydrogenase (GADPH) [17]. However, they typically failed to document whether the expression of the reference gene was stable. A stable expression of reference genes cannot necessarily be expected [18]. Specifically, studies in multiple cell types and tissues [19] including the heart [2] have found that exposure to β-AR agonists can down-regulate GAPDH expression; based on the elevated plasma catecholamine levels in CHF [10,11] this is likely to be of relevance for the mRNA data discussed below.

Cardiac α_1_-AR expression at the protein level has primarily been studied by radioligand binding and antibody-based techniques. While radioligands such as [^3^H]-prazosin or [^125^I]-BE 2254 (also known as [^125^I]-HEAT) have high specificity for α_1_-ARs, they do not discriminate between subtypes and, particularly in the case of [^3^H]-prazosin, are not very sensitive. In contrast, antibody-based approaches such as immunoblotting or immunohistochemistry potentially can be subtype-selective and be used to examine distribution within a tissue. However, multiple studies have demonstrated that the vast majority of commercially available α_1_-AR subtype antibodies lack target specificity when tested under stringent conditions [20,21,22]. These limitations need to be considered in the interpretation of the subsequently discussed studies.

Another important factor that might influence the results of different studies is the exact location of α_1_-ARs in the peripheral vasculature. It has been demonstrated that α_1_-AR receptors are located not only on smooth muscle cells, but, also on endothelial, adventitial and nerve cells [23,24]. This is a rather important finding that should be taken into account when interpreting results from expression and functional studies, since the total α_1_-AR mRNA and the actual role of specific α_1_-subtypes mediating vascular responses might differ depending on the specific cell on which they are expressed.

CHF can be caused by different pathologies, for instance by chronic myocardial ischemia, valvular heart disease or viruses, or can be idiopathic. As CHF in humans can be caused by a variety of pathophysiologic mechanisms, it is important to realize that subjects within an animal model typically are more homogeneous than patient cohorts; on the other hand, each model reflects a different cause and, potentially, a different pathophysiology. Among the many reported animal models of CHF, post-myocardial infarction [25,26,27], cardiac pacing causing tachycardiomyopathy [28], drug-induced [29,30] and hereditary models [31] have been applied to studies of cardiovascular α_1_-ARs. Finally, pathophysiology and regulation of α_1_-ARs may not only differ between patients and experimental models but also between species. While most animal studies related to cardiovascular α_1_-ARs in CHF were done with rats [2,26,32], guinea pigs [33,34], mice [27,29], hamster [35], cats [36] and dogs [37,38] have also been used.

## 4. Results

### 4.1. Study Selection

The literature search resulted in 1461 unique, original articles. G.K.-M. and O.P. independently screened the titles and abstracts of all the articles and excluded 1312 articles because of lack of relevance. After assessing the full text of the remaining 149 potentially relevant articles, and resolving any disagreements, the two reviewers decided to include 65 articles (Figure 1). Additional relevant studies that were retrieved from other articles were also included. This study selection process led to an inclusion of 33 unique reports on cardiac and 31 unique reports on vascular α_1_-AR. One study was reviewed for both cardiac and vascular α_1_-ARs leading to a total number of included studies of 65.

### 4.2. Cardiac α_1_-AR

#### 4.2.1. mRNA Expression

There are three α_1_-AR subtypes: α_1A_-, α_1B_- and α_1D_-ARs which have been identified in both animal and human hearts at the mRNA level [1,39,40]. All studies agree that α_1A_-AR is the most abundant subtype in the heart but the presence of α_1B_- and α_1D_-AR mRNA were also confirmed [1]. Although animal hearts express more α_1_-AR in the heart, human and rodent hearts are reported to locate α_1_-AR subtypes similarly, as α_1A_- and α_1B_-subtypes are predominantly found in myocardium whereas the α_1D_-subtype is expressed in coronary arteries [40,41]. Myagmar et al. claimed that α_1A_-ARs are present in a myocyte subset while α_1B_-ARs are in all cells [42]. Several studies reported changes in α_1_-AR subtype mRNA levels in CHF, but the results are heterogeneous (Table 1).

In human failing hearts, Fischer et al. demonstrated that mRNA of α_1A_- and α_1B_-ARs in the left ventricle was increased while no change in mRNA levels was observed in other cardiac regions [43]. Similarly, Jensen et al. reported that α_1A_ mRNA was increased in failing left ventricle LV and tended to increase in failing RV, but there was no change in α_1B_ and α_1D_ mRNA in any region [41]. On the other hand, Monto et al. did not confirm an altered expression of the three subtypes in human failing hearts; however, the relative abundance increased when considering the reduced β-AR expression [17]. They suggested that expression of α_1A_-ARs in the LV correlates with the left ventricular ejection fraction (LVEF) suggesting that this subtype may contribute to maintain cardiac inotropy in the failing heart. Moreover, a study in a rat coronary ligation model of CHF reported α_1D_-AR mRNA to be decreased; a numerical reduction of α_1A_- and α_1B_-AR mRNA did not reach statistical significance [2].

#### 4.2.2. Protein Expression

Cardiac α_1_-ARs were first identified at the protein level in rats by Williams and Lefkowitz [44] and thereafter confirmed in many species. Their quantitative expression at the protein level ranges from a relatively high expression in rodents to a low expression in humans, where their role appears considerably lower than that of β-ARs [1,39,40]. Despite numerous studies, data on α_1_-AR binding properties in human HF have remained inconsistent (Table 2). The density of α_1_-ARs was reported as increased [45,46,47,48], unaltered [41,45,49,50,51,52], or decreased [43,53].

However, even a lack of change in α_1_-AR protein expression may be biologically relevant for two reasons: Firstly, it is interesting to note that expression does not consistently decrease despite exposure to elevated plasma catecholamines [10,11]. However, studies outside the cardiovascular system have found that prolonged agonist exposure does not always lead to down-regulation of α_1_-AR subtypes and in some cases can even be associated with an up-regulation [16]. Secondly, an unaltered expression of α_1_-ARs in face of a consistently found down-regulation of β-ARs [1,39] implies a change in the relative role of the two adrenoceptor sub-families, which in its own right may have implications for the regulation of cardiac function [50,55].

A possible explanation for the controversial results on the regulation of cardiac α_1_-AR mRNA and protein is heterogeneity based on various etiologies of CHF. For instance, it was reported that α_1_-AR protein was increased in ischemic human cardiomyopathy but remained constant in dilated cardiomyopathy within the same study [45]. Similarly, the degree of HF may contribute to heterogeneous findings. While α_1_-AR density was not changed in a mild-to-moderate heart failure group, it was decreased in severely failing human hearts [52]. Finally, the change in α_1_-AR density may depend on the cardiac region under investigation: α_1A_- and α_1B_-AR protein were not changed in failing left ventricle, but α_1A_-AR binding was increased in right ventricle [41].

Studies in rats mainly found that α_1_-AR density remained constant after myocardial infarction [25,26,32,55,56], which is in line with the lack of a consistent change in human CHF (Table 2). However, an increase was also reported in a rat model of myocardial infarction [54]. Noradrenaline infusion, on the other hand, resulted in a decrease in α_1_-AR density in rats [30], demonstrating that cardiac α_1_-ARs are in principle susceptible to agonist-induced down-regulation.

There is also a substantial number of studies investigating the changes in cardiac α_1_-ARs using different animal models. Cardiomyopathic hamster [35] and dog [37] hearts were shown to have decreased α_1_-AR concentrations. In contrast, an increase in cardiac α_1_-AR density was demonstrated in a guinea pig model of HF caused by aortic constriction [33]. In addition, increase α_1_-AR densities were reported in cat [36] and guinea pig [34] ischemia models following coronary ligation.

#### 4.2.3. α_1_-AR Subtypes

For the interpretation of subtype expression data, there is a technical issue between α_1_-AR mRNA and protein levels. While mRNA of all α_1_-AR subtypes were detected in animal and human hearts, binding studies demonstrated the presence of only α_1A_- and α_1B_-subtypes, but not α_1D_-AR [39,41,57]. Lack of α_1D_-AR expression at the protein level has been shown in many tissues expressing corresponding mRNA [58].

Animal species do not only differ from humans by expressing more α_1_-AR in the heart but also by exhibiting a different subtype expression profile. For instance, rat [59] and mouse [58] hearts were reported to express α_1A_/α_1B_-AR in an approximate 20/80 ratio. Moreover, previous studies suggest that the three subtypes mediate different effects in the heart. α_1A_-AR has been reported to mediate positive inotropy in right ventricles of human hearts and this response has been found to be reduced only slightly in failing hearts. Thus, α_1A_-subtype-mediated inotropy is suggested to be significant source of inotropic support [60]. However, in nonfailing murine hearts, stimulation by the α_1A_-AR selective agonist, A61603 or the non-subtype-selective α_1_-AR agonist, phenylephrine mediated negative inotropic effects in right ventricle which was switched to positive inotropy in failing hearts [29]. On the other hand, the inotropic response mediated by α_1B_-subtype remained negative and unaltered. In the light of these findings, the authors suggested that α_1A_-subtype may play a role in improving contraction in RV failure and might be a therapeutic target. Another study using transgenic mice with heart-specific overexpression of α_1A_-AR displayed enhanced inotropy after myocardial infarction [61]. Moreover, α_1D_-AR mRNA decreased in rats after coronary ligation; however, this study was inconclusive because there was a non-significant trend for reduced mRNA of α_1A_- and α_1B_-ARs [2]. Thus, the authors proposed that a decrease in α_1D_-AR might be a protective response of the failing heart. However, the relevance of these data remains unclear because studies in several mouse tissues have demonstrated that many tissues express α_1D_-AR at the mRNA but not at the protein level [58].

Patients who receive left ventricular assist device to unload the failing heart and improve cardiac function were shown to have a redistribution of α_1A_-AR from a peri-myocytic to an intra-myocytic location along with a rise in the overall density of receptors [62]. On the other hand, α_1A_-AR subtype was decreased in human dilated cardiomyopathy and this result was concluded to be consistent with the theory that α_1A_-AR agonism may be cardioprotective [53]. Fischer et al. demonstrated that α_1A_- and α_1B_-AR mRNA levels in left ventricle of human failing hearts were increased, although α_1A_-, α_1B_- and α_1D_-AR binding sites were decreased [43]. They suggested that a decrease of α_1B_-AR can have cardioprotective effects while a decrease of α_1A_-AR is more likely the result of impaired heart function.

#### 4.2.4. Signaling Pathways Involved

A schematic representation of α_1_-AR signal transduction is shown in Figure 2. The prototypical signaling pathway of α_1_-ARs involves coupling to G_q_ with subsequent activation of phospholipase C (PLC), which releases Ca^2+^ from intracellular stores and activates protein kinase C (PKC) [63]. However, cardiac α_1_-ARs can also couple to G_i_ [5,7], G_12/13_ [64], even G_s_ proteins [65] and G_h_ which is a GTP-binding protein with Transglutaminase activity [66] and to pathways such as inhibition of adenylyl cyclase [67], modulation of Ca^2+^, K^+^ and Cl^-^ channel activity [7,68,69], ras activity [70] or activation of various mitogen activated protein kinases [71]. Most of these pathways have also been studied in human CHF or CHF animal models [72].

Activation of G_q_-signaling pathways mediates the development of hypertrophy and the transition to HF [73]. Overexpression of G_q_ was demonstrated to result in marked hypertrophy [74] and HF [75]. Moreover, inhibition of G_q_-mediated signaling was shown to inhibit cardiac dysfunction [76,77]. On the other hand, multiple studies in humans and several animal species have shown that HF is accompanied by an increased expression of G_i_ at the mRNA and protein level, whereas a reduced expression of G_s_ has been found less consistently [78]. Reduced GTP binding with Transglutaminase activity of Gα_h_ were found in human HF [45]. α_1_-AR coupling to a GTP binding protein was detected only in pathological, but not in healthy human hearts [47].

α_1_-AR-mediated phosphoinositol turnover is not changed in both animal and human studies when accompanied by an unaltered receptor density in human [50] and rat [56] HF. On the other hand, in a rat model of myocardial infarction, an increase in phosphoinositol turnover was reported along with an increase in G-protein content and an upregulation of α-skeletal actin which suggests an involvement of α_1_-ARs in the reactive hypertrophic response [26].

α_1_-AR stimulation increases both the intracellular Ca^2+^ concentration, which activates muscle contraction, but also the Ca^2+^ sensitivity of cardiomyocytes. α_1_-AR-mediated Ca^2+^-transients and phosphorylation of cardiac myosin regulatory light chain were increased in a mouse right ventricular HF model [29]. The increase in Ca^2+^ sensitivity expected from these observations was reported in a canine HF model, apparently occurring via the G_q_-RhoA-ROCK signaling pathway [38]. Similarly, the switch from negative to positive inotropy observed in mouse failing right ventricles (see below) was claimed to be mediated through a pathway involving increased myofilament Ca^2+^ sensitivity [27].

#### 4.2.5. Cell and Tissue Responses

α_1_-AR stimulation induces a positive inotropic effect in isolated cardiac preparations such as atria, left and right ventricles (Table 3) [25,55,56,60]. The positive inotropic effect of α_1_-AR stimulation involves multiple subtypes, but mostly occurs via α_1B_-AR in rodents; when these become inactivated in vitro, the α_1A_-component becomes stronger and compensates for the loss of α_1B_ response resulting in overall maintained contractile responses [79]. However, depending on experimental conditions α_1_-AR stimulation can also have negative inotropic effects (Table 3) [29]. Moreover, stimulation of cardiac α_1_-AR can increase heart rate [56], but systemic stimulation can reduce heart rate secondary to an increase in total peripheral resistance and activation of the baroreflex [80], which can mask direct chronotropic effects of α_1_-AR stimulation.

The chronotropic and inotropic responses to the α_1_-AR agonist phenylephrine were decreased in isolated right and left atria of rats with myocardial infarction [56]. As this was not accompanied by alterations of receptor density or phosphoinositol turnover, this may reflect changes of post-receptor signaling (see above). In contrast, several studies in isolated papillary muscle from humans [49,51] or rats [25,55] reported unaltered inotropic responses in CHF, which was in line with unaltered receptor density reported in these studies. The positive inotropic response to α_1A_-AR stimulation in human right ventricle also exhibited little change in HF in two studies [60,81]. In contrast, the phenylephrine-induced positive inotropic response in human CHF was abolished in another study [46]. In mouse models, negative inotropic effect observed in control animals was switched to positive inotropy in the failing right ventricles [29]. However PE-mediated positive inotropy in LV of nonfailing hearts remained unchanged in failing mouse hearts after myocardial infarction [27]. Cardiomyopathic hearts of Syrian hamsters were shown to have increased inotropy accompanied by increased Ca^2+^ levels and pertussis toxin-sensitive G-proteins [82]. Thus, with few exceptions, most studies reported little change in inotropic responses in the failing heart across species and models. This may nonetheless be biologically relevant because it would indicate an α_1_/β-AR balance in light of the consistently found reduction in β-adrenergic function in CHF [50,55].

Moreover, there are opposing views on the role of α_1_-AR subtypes on hypertrophy. While some suggest that α_1A_-AR were found to mediate hypertrophy development in neonatal rat cardiomyocytes in vitro [8], transgenic overexpression of α_1B_-AR caused a hypertrophic response resulting in HF, while α_1A_-AR overexpression did not affect hypertrophy and did not hasten development of HF [29,83].

### 4.3. Vascular α_1_-AR

#### 4.3.1. mRNA Expression

Multiple studies have explored the expression of α_1_-AR subtype mRNA in arteries and veins of various species including rat [84,85,86,87,88], rabbit [89,90,91] and human [89,92,93,94]. It appears from these and many other studies, that all three α_1_-AR subtypes are expressed in the vasculature of these species in a species- and vascular bed-dependent manner. This conclusion is in line with studies on the role of α_1_-AR subtypes in functional studies [3]. Our search did not identify any animal studies reporting on the regulation of vascular α_1_-AR subtype mRNA in CHF. However, it is likely that such regulation occurs because CHF is characterized by increased circulating catecholamine levels and the endogenous agonist noradrenaline has been shown to transiently down-regulate α_1_-AR mRNA in rabbit aortic smooth muscle cells whereas the down-regulation of α_1_-AR protein was longer lasting (at least 96 h) [95]. Moreover, CHF can be associated with hypoperfusion and hypoxia of tissues and vascular α_1_-AR mRNA expression can be regulated by hypoxia [96]. Vascular α_1_-AR mRNA expression is also regulated by age [97]. HF did not appear to affect the total levels of α_1_-AR mRNA in human coronary arteries [98]. On the contrary, the use of β-blockers was associated with a significant decrease in both α_1D_ and total α_1_-AR mRNA levels. α_1D_-AR appeared to be the principal receptor in human epicardial coronary arteries, which resembles the mouse coronary arteries [99,100].

#### 4.3.2. Protein Expression

Quantitative, ligand-binding studies have been performed in order to assess the α_1_-AR distribution in the various vascular beds of different species. Existing discrepancies might be due to interspecies differences, variant α_1_-AR subtype distribution in the peripheral vasculature, differences in models of HF, for example pacing-induced HF versus myocardial infarction, or even due to poorly reproducible results.

Such studies have been performed in animals and humans without coexisting diseases in order to assess the physiological α_1_-ARs at the protein level. In humans a considerable difference in α_1_-AR protein expression has been demonstrated between somatic (mammary artery) and splanchnic arteries [97]. However, this was not true for veins, in which the α_1_-AR density seemed to be independent from differences in vessel diameter. This might be explained by the importance of splanchnic circulation, especially during a situation of hypovolemic shock. Sympathetic stimulation of α_1_-ARs in those vascular territories leads to maintenance of blood flow and O_2_ delivery to vital organs. On the other hand, constriction of conduit arteries and veins contribute to the preservation of blood pressure and cardiac afterload. In human epicardial coronary arteries 8.7 fmol/mg protein of total α_1_-ARs were found, of which 75% constituted of the α_1D_ subtype, almost twice as much as in the left ventricle [41,98].

Considering that progression of HF is associated with a decrease in β-AR protein in the heart, one could argue that similar findings would be expected for α_1_-ARs. On the contrary, α_1_-AR density has been reported to be increased, yet not statistically significant, in studies with myocardial infarction-induced CHF in rats [101,102]. However, the lack of statistical significance of this change leads to rather inconclusive results.

#### 4.3.3. α1-AR Subtypes

All three subtypes have been identified in the peripheral vasculature of various species, however the subtype distribution varies between animals and humans, between different vascular beds and with age. For instance, α_1B_ and α_1D_ subtype mRNAs have been detected in human aorta [93] and are considered to mediate vascular constriction. A controversy exists in the rat aorta since some studies have identified only the subtype α_1D_ [103,104,105], while others report both α_1B_ and α_1D_-ARs [101]. The α_1B_ subtype has been recognized to mediate vasoconstriction after sympathetic stimulation in the mouse, rabbit and dog aorta [106,107,108]. Similarly, mesenteric responses in humans have been found to be mediated by all three α_1_-AR subtypes [109], whereas in rats mesenteric vascular constriction is mediated by α_1D_ and α_1B_ subtypes in superior mesenteric artery and the mesenteric resistance arteries respectively [101,104]. In the small mesenteric arteries the subtype α_1A_ has also been identified [101,110]. The same receptor was found to mediate contractile responses in rat small renal arteries [111].

Similarities in vascular tone regulation between different species also exist. As it was mentioned before, α_1D_ is recognized as the main subtype that mediate coronary contraction in humans [98] and mice [99,100] and the same has also been reported in rats [105]. In this latter study, it was additionally shown that CHF did not modify the function of α_1D_-AR, neither in the carotid artery nor in the aorta, whereas the α_1D_-AR-mediated vasoconstriction was enhanced in older animals.

Furthermore, blood pressure regulation in mice is considered to be achieved via the stimulation of all three subtypes. This is confirmed in knockout mice studies in which deletion of any subtype results in a decrease in resting blood pressure [106,112,113].

The heterogeneous α_1_-AR population in the various vascular beds might be modified depending on the disease state. In support of this, antagonist studies have explicitly displayed an alteration of the antagonism against α_1_-ARs before and after the onset of CHF [114,115,116,117] and showed changes from competitive to non-competitive antagonism, or the other way around, which, however, was also related to the specific agonist and vessel studied. Distinct patterns of antagonism have been identified in different disease states and in different vessels. The specific subtype populations, however, and their involvement in the vascular responses in different disease settings are still to be elucidated.

#### 4.3.4. Signaling Pathways Involved

Changes in α_1_-AR responsiveness due to chronic exposure of the peripheral vasculature to the endogenously released catecholamines might also derive from alterations in signal transduction. These changes might be due to modification of Ca^2+^ sensitivity and/or other changes in the signaling pathways involved. For example, the continuous exposure of smooth muscle cells from rabbit aorta to α_1_-AR agonists led to a major loss in sensitivity, which did not seem to be due to down-regulation of α_1_-ARs, but rather to blunting of phosphatidylinositol turnover [118].

Disease progression might also modify the dependency of α_1_-AR-mediated vascular contraction on extracellular Ca^2+^ which seems to increase in end-stage HF [119,120]. More precisely, the existence of a high affinity component of the α_1_-AR-mediated responses was identified which is inhibitable by a PKC inhibitor, polymyxin B, and which becomes apparent with the progression of HF. Interestingly, in this study, the dorsal pedal artery presented lower dependency on PKC activation than the saphenous vein, which underlines the existing differences between separate vascular beds [120]. Likewise, after investigating the α_1_-AR in mesenteric resistance arteries from CHF rats, it was confirmed that Ca^2+^ plays a more crucial role in the vascular contractility as CHF develops [121]. Furthermore, it has been demonstrated that the diminished contractile response of rat mesenteric resistance arteries to phenylephrine and Ca^2+^ is not due to changes in Ca^2+^ sensitivity, nor in intracellular Ca^2+^ release, but rather to changes in the α_1_-AR-triggered Ca^2+^ influx [102].

Other studies, however, have identified a modification in the sensitivity of α_1_-AR vasoconstriction to Ca^2+^. For instance, in CHF rats an increased responsiveness of CHF femoral resistance arteries was identified which appeared to be the result of first, a RhoA/Rho kinase-depended increase in sensitivity to Ca^2+^ and second, of the IP3-induced increase in intracellular Ca^2+^ [122]. On the contrary, this arterial sensitivity to Ca^2+^ was reported to be decreased in femoral resistance and muscle resistance arteries of CHF rats, which presented as a dissociation between the intracellular Ca^2+^ and the generated wall tension [123].

#### 4.3.5. Cell and Tissue Responses

α_1_-ARs constitute an important homeostatic mechanism that contributes to the maintenance of peripheral vascular resistance at normal conditions and also in response to shock. Nevertheless, α_1_-AR regulation and function might alter due to disease. Even though HF is characterized by enhanced vascular resistance [124], there is little, and conflicting evidence to date concerning the vascular contractile responses to α_1_-AR stimulation in HF vessels.

The chronic sympathetic stimulation during HF might result in the desensitization of vascular α_1_-AR and consequently, in the failure of the contractile apparatus to function properly. Attenuation of α_1_-AR responsiveness after continuous exposure to catecholamines has been validated in various animal studies [125,126] but also in vitro [127,128]. However, despite the fact that several studies have tried to assess the α_1_-AR mediated vascular responses in both animals and humans with HF, the findings are not unanimous (Table 4). Multiple studies have shown an increase in vascular responsiveness to α_1_-AR stimulation in animals [114,115,116,117,119,120,122,129,130,131]. On the contrary, others have reported desensitization of α_1_-ARs in the peripheral vasculature in animals [102,121,132,133,134,135,136] and in humans [137,138,139,140], while numerous studies failed to demonstrate any alterations of the α_1_-AR responsiveness in vessels from animals [105,123,133,141,142,143] and humans [144].

Inconsistency in findings might originate from differences in the anatomy and function of distinct vascular beds. Additionally, differences in CHF model and disease state might also influence the results of different studies. For example, in some studies it was demonstrated that the saphenous vein of CHF dogs was at all times more sensitive than the dorsal pedal artery [114,115,116,120]. This might be associated to a heterogeneous smooth muscle function between the artery and the vein, but, also, to differences in receptor population in the canine vasculature [148]. In addition, differences in vascular responsiveness between different vascular beds were also identified in rat models of HF [123,130]. Moreover, differences have also been identified between the various arteries studied. For example, in similar models of pacing-induced CHF in dogs, some studies have demonstrated an increased responsiveness to α_1_-AR stimulation in the dorsal pedal artery and the contralateral saphenous vein [114,115,116,119], while others have identified a desensitization of α_1_-ARs in coronary arteries of CHF dogs [132].

Other important parameters that influence the peripheral vascular responses are the integrity and physiological function of endothelium. More specifically, the endothelium-induced vasodilation counteracts the contraction via α_1_-ARs. Therefore, the malfunction of the endothelium might lead to the absence of this inhibitory function in CHF vessels [143] and result in an enhancement of vascular contraction. Increased vascular responses to noradrenaline were reported in the intact thoracic aorta from HF rats compared to controls, however the actual vascular α_1_-AR responsiveness was diminished when the endothelium was removed [146]. On the other hand, an increase in endothelial function as was observed in other studies might oppose the α_1_-AR-mediated vascular contractions [131,132]. Others, however, did not demonstrate a difference in vascular responses between intact and denuded vessels [123].

Even though the vascular responsiveness in patients with HF has not been widely investigated, many studies have yielded data that might have important clinical implications. More precisely, Schwinn et al. [145] reported on the vascular responsiveness of HF patients that underwent coronary artery revascularization. The results from this study support that left ventricular dysfunction and more precisely, left ventricular ejection fraction equal to or lower than 40% is associated with a diminished vascular responsiveness during fentanyl anesthesia in this patient population. In contrast to more recent reports that identify cardiac surgery and cardiopulmonary bypass as predisposing factors for an attenuated vascular responsiveness [13,14,149], Schwinn et al. [145] reported that phenylephrine-induced changes in systemic vascular resistance were increased during cardiopulmonary bypass and aortic cross-clamp in both HF and control subjects. This vascular responsiveness is of interest in HF patients that undergo cardiac surgery, since this patient population exhibits an increased risk for post-operative vasoplegia [150]. This complication is characterized by low systemic vascular resistance, severe hypotension, normal or increased cardiac output and a blunted or absent response to the administration of vasopressors [151]. The diminished sensitivity of HF patients is thought to originate from the continuous, chronic exposure of these patients’ vascular α_1_-ARs to catecholamines which leads to desensitization and/or downregulation of those receptors. If vasoplegia is not promptly treated, the resulted tissue malperfusion might lead to end-organ dysfunction and multiple organ failure. Therefore, this post-operative complication significantly increases the risk of morbidity and mortality [152,153]. Moreover, another study identified a maladapted sympathetic response of the skeletal muscle vasculature to exercise that was characterized by sustained vascular constriction [140]. This can have serious implications on normal blood flow during physical movement and might hamper the patient’s daily life. Furthermore, the apparent up-regulation of α_1_-AR responsiveness after chronic carvedilol treatment in HF patients [147] and the lack of change after chronic angiotensin-converting enzyme inhibitors [142] might yield important consequences regarding HF therapy.

## 5. Discussion

Understanding the role of α_1_-ARs is particularly important in HF because this condition is associated with high adrenergic activation leading to down-regulation and desensitization of cardiac β-ARs. α_1_-ARs have been identified in both animal and human hearts and vasculature with a high expression in rodents but a much lower abundance in humans. Thus, caution should be applied while extrapolating data from animals to humans.

The α_1A_-AR is the most abundant subtype in the human heart and α_1B_-AR in the rodent heart, while α_1D_-ARs are mainly found in coronary arteries (Figure 3). However, studies on a change in the expression of the subtypes in failing hearts are inconsistent, which may indicate a relative α_1_/β-AR imbalance in light of the well-known reduction in β-adrenergic function in CHF.

Another important point is that, while in heart α_1A_- and α_1B_-ARs dominate, all three subtypes are expressed in the vasculature and their relative expression and physiological role appear to differ between vascular beds and species (Figure 3) [3]. This varied expression of α_1_-AR has been reported to correlate with the adrenergic innervation in peripheral vessels [154]. Moreover, studies with transfected receptors suggest that the three α_1_-AR subtypes undergo differential regulation upon exposure to agonists [16]. Certainly, those discrepancies also affect the way in which blood pressure is regulated depending on which α_1_-AR subtypes play the predominant role. For that reason, the differences in subtype distribution might also have influenced studies that have investigated the function of α_1_-ARs in vivo or in vitro, leading to conflicting results.

α_1_-ARs can couple to multiple intracellular signal transduction pathways in the heart. Under most conditions α_1_-AR signal transduction results in enhanced inotropy. Although data on their role in HF is limited, there is evidence indicating that they mediate enhanced positive inotropy, as well as physiological hypertrophy and decreased apoptosis [46]. α1-AR occupancy was estimated as 10% in failing human heart [155]. Thus, activation of unoccupied receptors with an exogenous ligand is suggested to be cardioprotective [46]. Clinical trials such as ALLHAT in hypertension and V-HeFT in CHF reported that treatment with α_1_-AR antagonists was associated with greater mortality and apoptosis than with other treatments [156]. Moreover, double knockout of α_1A_- and α_1B_-AR was shown to augment apoptosis, worsen cardiomyopathy and reduce survival [157]. Taken together, cardiac α_1_-ARs appear to mediate cardioprotective effects. Data on the regulation of vascular α_1_-ARs in HF are also conflicting. This is unfortunate because of their role in controlling total peripheral resistance and the role of afterload reductions in the treatment of patients with HF. However, clinical studies in hypertensive patients have shown that α_1_-AR antagonism provides less protection from HF than other classes of anti-hypertensive drugs [156]. More specifically, the non-selective α-blocker doxazosin was associated with significantly higher risk of HF compared with the other treatment arms of this trial. In the absence of a placebo arm, it remained unclear whether this reflects an adverse effect of doxazosin or less protection against hypertension-induced HF. The results from the V-HeFT trial [158] further support the detrimental effects of α-blocking therapy-, since prazosin use gave a signal for possibly increased mortality that failed to reach statistical significance. The reason for those results could possibly be explained by the findings from knock-out mice studies which confirm the importance of α_1_-AR signaling for the adaptational mechanisms of cardiomyocytes in disease settings [157,159].

In contrast to those studies that investigated the effect of selective α-blockade and discouraged their widespread use, the mixed β- and α_1_-AR antagonist carvedilol was demonstrated to be more beneficial than the β_1_-selective antagonist metoprolol in diminishing the risk of mortality [155]. It could, therefore, be postulated that the favorable effects of carvedilol are to some extent related to its α-blocking properties. Nevertheless, the combination of an α- with a β-blocker for long-term treatment in chronic CHF patients, did not yield better clinical outcomes than β-blocker therapy alone [160], which highlights the fact that such a combined therapy does not generate satisfactory outcomes similar to that of carvedilol.

The exact reason for those discrepancies in effectiveness still remains unclear. One explanation could be that carvedilol’s α-blocking properties are less potent compared to both its innate β-blocking feature and to the α-blockade produced by a specific α-AR antagonist like prazosin. This is in fact supported by the results of a small trial in which carvedilol did not antagonize phenylephrine induced responses [161]. Another hypothesis is that the existence of distinct pharmacologic properties and mechanisms of action between the various α- and β-blockers could also lead to the occurrence of different side effects. Based on this assumption, one could possibly explain the contradictory findings from studies with carvedilol versus studies investigating the combination of an α- with a β-blocker. Last but not least, carvedilol use leads to pharmacologic tolerance after long term use [162]. This implies that its beneficial effects at long-term use derive mostly from its β-blocking properties rather than antagonizing α_1_-ARs.

In brief, one could state that α_1_-AR agonism could potentially be a valid treatment option for HF patients that would stimulate innate adaptive signaling procedures and would eventually illustrate its cardioprotective features. However, caution is required, since α_1_ agonism could also have detrimental effects by increasing peripheral vascular resistance leading to an increased cardiac afterload and by inducing coronary vasoconstriction in those patients. Therefore, future research is necessary which should focus on compartment and/or subtype selective treatment in HF patients and which will investigate in more detail the treatment modalities of α_1_ agonism.

On the other hand, carvedilol seems promising for the treatment of HF patients and might be an option for the pretreatment of HF patients that will undergo cardiac surgery. Short term α_1_ antagonism with carvedilol treatment could shield α_1-_ARs against continuous catecholamine exposure in the preoperative period. This could perhaps increase the sensitivity of those receptors to the administration of catecholamines for the treatment of vasoplegia during the postoperative course [147].

In conclusion, more research is required to further understand the complex mechanisms of α_1_ adrenergic function and adaptation in HF. Taking into account the existing differences between animals and humans regarding the expression of α_1_-ARs, we should carefully consider our next steps and we should promote additional research which would focus more on human tissue (heart and vessels). Moreover, a cardioselective approach in the case of α_1_ agonism appears to be crucial for the exploitation of only its advantageous characteristics by avoiding potential vascular side effects. A subtype approach could also be beneficial but further studies are necessary in order to gain more insight in such treatment techniques. Finally, in view of the positive outcomes from clinical studies with carvedilol, more extensive investigation is to be expected so as to better comprehend the pharmacologic properties and mechanism of action of this agent. It is, therefore, essential that we expand our knowledge, and we apprehend current inconsistencies in order to pave the way to safer and more effective therapies.

## Figures and Tables

**Figure 1 cells-09-02412-f001:**
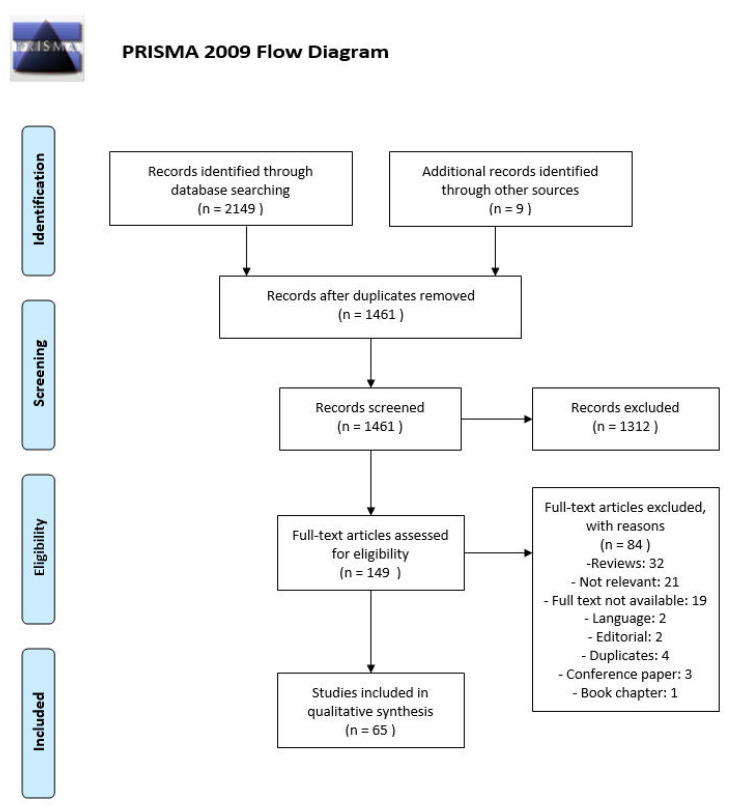
Preferred Reporting Items for Systematic Reviews and Meta-Analyses (PRISMA) study flow chart.

**Figure 2 cells-09-02412-f002:**
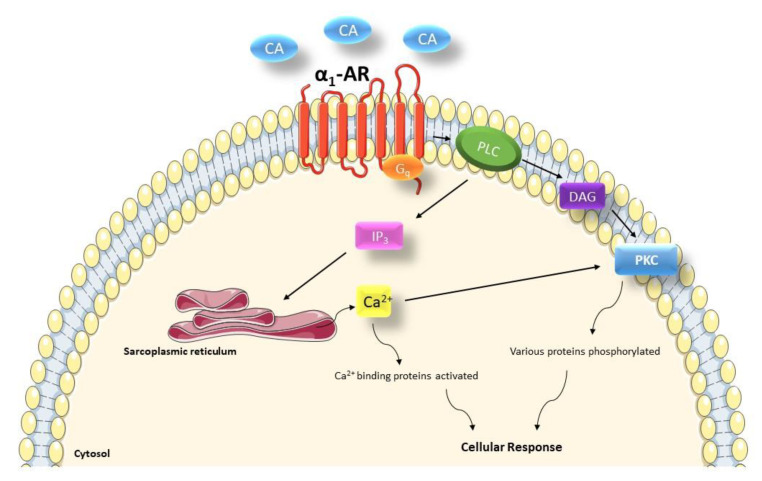
The prototypical signaling pathway of α_1_-ARs. CA: catecholamines, DAG: diacylglycerol, IP3: inositol triphosphate, PKC: protein kinase C, PLC: phospholipase C.

**Figure 3 cells-09-02412-f003:**
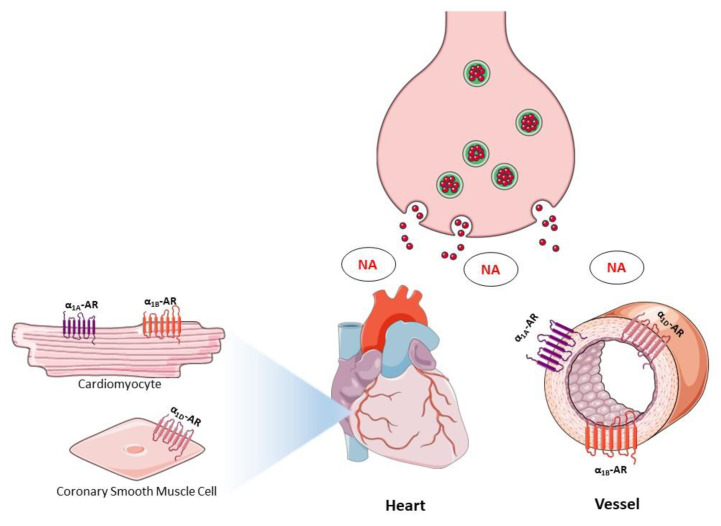
Differential expressions of α_1_-ARs in the cardiovascular system. NA: noradrenaline.

**Table 1 cells-09-02412-t001:** Summary of the changes in cardiac mRNA studies for total α_1_-adrenoceptors (AR) and their subtypes in chronic heart failure.

mRNA	Species	Tissue	α_1A_-AR		α_1B_-AR	α_1D_-AR	Total α_1_-AR
Fischer et al. [43]	human	LV, RV, LA, RA	↑		↑	-	n/a
Jensen et al. [41]	human	LV and RV	↑		X	X	X
Monto et al. [17]	human	LV and RV	X		X	X	n/a
Brattelid et al. [2]	rat	LV	X		X	↓	n/a

↑: increased; ↓: decreased; X: unchanged; n/a: no data available; LV, left ventricle; RV, right ventricle; LA, left atrium; RA, right atrium.

**Table 2 cells-09-02412-t002:** Summary of cardiac α_1_-adrenoceptor (AR) protein expression in chronic heart failure.

Binding	Species	Tissue	α_1A_-AR	α_1B_-AR	α_1D_-AR	Total α_1_-AR
Böhm et al. [49]	human	papillary	n/a	n/a	n/a	X
Bristow et al. [50]	human	LV and RV	n/a	n/a	n/a	X
Corr et al. [36]	cat	ventricle	n/a	n/a	n/a	↑
Dixon et al. [54]	rat	LV	n/a	n/a	n/a	↑
Erdmann and Böhm [51]	human	papillary	n/a	n/a	n/a	X
Fischer et al. [43]	human	LV, RV, LA, RA	↓	↓	↓	n/a
Gopalakrishnan et al. [32]	rat	LV and RV	n/a	n/a	n/a	X
Hwang et al. [45]	human	explanted hearts	n/a	n/a	n/a	↑ (in ICM) X (in DCM)
Jensen et al. [41]	human	LV and RV	X (↑ in RV)	X	undetected	X
Karliner et al. [33]	GP	whole heart	n/a	n/a	n/a	↑
Limas et al. [52]	human	LV and RV	n/a	n/a	n/a	↓ (in severe HF) X (in mild)
Litwin et al. [25]	rat	LV	n/a	n/a	n/a	X
Maisel et al. [34]	GP	LV	n/a	n/a	n/a	
Meggs et al. [26]	rat	LV and RV	n/a	n/a	n/a	X
Re et al. [37]	dog	LV, RV, LA, RA	n/a	n/a	n/a	↓
Shi et al. [53]	human	LV	↓	X	n/a	↓
Sjaastad et al. [55]	rat	LV and RV	n/a	n/a	n/a	X
Steinfath et al. [46]	human	LV	n/a	n/a	n/a	↑
Watanebe et al. [35]	hamster	whole heart	n/a	n/a	n/a	Slight↓
Zhao et al. [30]	rat	RA and LV	n/a	n/a	n/a	↓
Rowley et al. [56]	rat	atria	n/a	n/a	n/a	X
Vago et al. [47]	human	LV	n/a	n/a	n/a	↑

GP: guinea pig, ↑: increased; ↓: decreased; X: unchanged; n/a: no data available; LV, left ventricle; RV, right ventricle; LA, left atrium; RA, right atrium.

**Table 3 cells-09-02412-t003:** Summary of changes in cardiac α_1_-adrenoceptor (AR)-mediated responses in chronic heart failure.

	Species	Preparation	α_1_-AR-mediated Responses
Böhm et al. [49]	human	papillary	X
Erdmann and Böhm [51]	human	papillary	X
Litwin et al. [25]	rat	papillary	X
Sjaastad et al. [55]	rat	papillary	X
Cowley et al. [29]	mouse	trabecula	α_1A_–med↑ α_1B_–med X
Janssen et al. [60]	human	trabecula	slight ↓
Steinfath et al. [46]	human	trabecula	↓
Wang et al. [27]	mouse	trabecula	in RV switch from NIE to PIE in LV X
Skomedal et al. [81]	human	trabecula	PIE
Rowley et al. [56]	rat	atria	↓

↑: increased; ↓: decreased; X: unchanged; RV: rigt ventricle; LV: left ventricle; NIE: negative inotropic effect; PIE: positive inotropic effect.

**Table 4 cells-09-02412-t004:** Summary of changes in vascular α_1_-adrenoceptor (AR)-mediated responses in chronic heart failure.

	Species	Preparation/Parameter Studied	α_1_-AR-Mediated Responses
Goldsmith et al. [137]	human	In vivo hemodynamic responses	↓
Wilson et al. [141]	dog	In vivo hindlimb vascular responses	X
Schwinn et al. [145]	human	In vivo BP response	X (awake)/ ↓ (on fentanyl anesthesia)
Borow et al. [138]	human	In vivo BP response	↓
Forster et al. [114]	dog	DPA and SV	↑
Forster et al. [115]	dog	DPA and SV	↑
Main et al. [132]	dog	LAD and LCX	↓
Forster et al. [119]	dog	DPA and SV	↑
Angus et al. [139]	human	Small arteries from gluteal skin biopsy	↓
Townsley et al. [129]	dog	Pulmonary arterial and venous pressure response	↑
Indolfi et al. [144]	human	In-vivo forearm blood flow response	X
Teerlink et al. [146]	rat	Thoracic aorta	↑ intact vessels/ ↓ denuded vessels
Forster et al. [120]	dog	DPA and SV	↑ significant only in SV
Bergdahl et al. [130]	rat	Basilar, femoral, and renal artery and iliac vein	↑ arteries (NS) ↓ vein
Forster et al. [116]	dog	DPA and SV	↑
Stassen et al. [102]	rat	MSA	↓
Stassen et al. [133]	rat	Thoracic aorta, coronary arteries, MSA	X thoracic aorta/ ↑ coronary arteries (NS)/↓ MSA
Mulder et al. [142]	rat	Abdominal aorta, Femoral artery, MSA	X
Le Tran et al. [117]	dog	DPA and SV	↑ (significant only for NA in DPA)
Martinez et al. [105]	rat	Aorta, Carotid artery	X
McMillon et al. [143]	dog	Intrapulmonary bronchial vessels	X
Feng et al. [134]	rat	In vivo BP response	↓
Bergdahl et al. [121]	rat	MSA	↓
Ahmadiasl et al. [135]	rabbit	Thoracic aorta, left renal artery and vein, lateral saphenous artery and vein, vena cava	↓ significant only in vena cava
Tamagawa et al. [131]	dog	LAD/ In vivo coronary pressure-flow relationship response	↑
Trautner et al. [123]	rat	Femoral artery (2nd order side branches) and muscle resistance arteries	X
Koida et al. [122]	rat	Femoral resistance artery (small branch)	↑
Van Tassel et al. [147]	human	In vivo BP response	↑ with up-titration of carvedilol
Ramchandra et al. [136]	sheep	Renal vessels/In vivo renal vasoconstrictor response	↓
Barrett-O’Keefe et al. [140]	human	CFA and CFV/In vivo leg blood flow response	↓at rest

BP: blood pressure, CFA: common femoral artery, CFV: common femoral vein, DPA: dorsal pedal artery, LAD: left anterior descending coronary artery, LCX: left circumflex coronary artery, MSA: mesenteric small artery, NS: not significant, SV: saphenous vein, ↑: increase, ↓: decrease, X: similar.

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
