# Peer review of "Cardiac and Vascular α1-Adrenoceptors in Congestive Heart Failure: A Systematic Review"

_cells, 2020, doi:10.3390/cells9112412_

Round 1
Reviewer 1 Report
This is a good topic and will be interesting to readers.
Though alpha-receptor is not as important as beta-receptor in the heart, it still plays an important role.
The authors reviewed references that are pertinent to the topic. The review described the mRNA and protein changes under stress from the references. Alpha-receptor is very important in vessel system. The vessel system in the heart is not a small part which can be ignored when doing mRNA and protein analyses. I am wondering whether the references reviewed measure the expression of the whole heart or in separated cell types such as myocytes, smooth muscle cells, or endothelial cells. Are there any reports who has isolated myocytes and smooth muscle cells from the heart to measure the changes in expression?
The authors have mentioned that GAPDH as reference gene in these studies has limitations. What are other reference genes used for this purpose in similar studies? Are they better or same compared to GAPDH?
Table one: the legend is not clear. Under what situation did they report the changes in the table?
Overall, it is a good review article.
Author Response
“I am wondering whether the references reviewed measure the expression of the whole heart or in separated cell types such as myocytes, smooth muscle cells, or endothelial cells. Are there any reports who has isolated myocytes and smooth muscle cells from the heart to measure the changes in expression?”
There are studies showing that α1A- and α1B- subtypes are located in cardiac myocytes, whereas α1D- subtype is present in coronary smooth muscle cells and α1B- subtype in coronary endothelial cells. In contrast, no subtype is expressed in cardiac fibroblasts. However, the studies reviewed in our manuscript, demonstrating how heart failure alters α1-ARs, report changes in the whole heart or from different sections of the heart (e.g., l. 127, 133, 163 and 167 to name but a few) but not from separated cell types. We now specifically mention what regions were studied in Tables 1 and 2.
“The authors have mentioned that GAPDH as reference gene in these studies has limitations. What are other reference genes used for this purpose in similar studies? Are they better or same compared to GAPDH?”
Besides GAPDH, β-actin, hypoxanthine phosphoribosyltransferase (HPRT), and β-2-microglobulin are among commonly used reference genes. Each reference gene may have advantages to others according to the experimental conditions. Thus, suitability for a given reference gene must be tested for each experimental setting.
“Table one: the legend is not clear. Under what situation did they report the changes in the table?”
The information that the table summarizes the changes observed in CHF has been added.
Reviewer 2 Report
This is the systematic review article which investigated the role of alfa1 adrenergic receptors in chronic heart failure. This paper thoroughly explored the previous studies of animal and human samples regarding the distribution, function of alfa1-AR and its subtypes. As a conclusion, the contribution and the potential role of alfa1AR on the pathology are controversial and remains elusive, but the data shown in this paper are well summarized and valuable for the scientists.
Overall, the contents in this paper looks reasonable. The Reviewer just felt that the text was lengthy and should be condensed a little bit, especially sections 4.2-4.3.
Author Response
“The Reviewer just felt that the text was lengthy and should be condensed a little bit, especially sections 4.2-4.3.”
While a comprehensive review can sometimes feel to be lengthy, this is at least partly unavoidable based on covering all applicable studies. Nonetheless, we have followed your suggestion and shortened in several places.
Reviewer 3 Report
The growing health problem is heart failure in our population. Activation of the sympathetic system influence all adrenergic receptors, and thus also α1 adrenergic receptors. Pharmacological intervention of substances acting on all adrenergic receptors in the right combination can alleviate the consequences of heart failure. This systematic review of the involvement of α1 adrenergic receptors in congestive heart failure in the heart and vascular system is beneficial in this context.
I have one comment on the overall concept of the text. This manuscript includes publications since 1984. It is very commendable. On the other hand, some of the information used may already be outdated. For example, the information "In contrast, α1-AR coupling to a GTP binding protein was only detected in pathological, but not in healthy human hearts" taken from a publication from 1989 would deserve more attention. The association of α1 adrenergic receptors with G protein has been based on binding studies only.
However, this manuscript contains many technical flaws.
- The text states that it is based on 32 publications on the heart and 32 on blood vessels. In contrast, in Figure 1, the total number of used publications is 65.
- Figure naming should be below the Figure. Figures lack any explanatory text.
- PKC can be activated by calcium ions too, in Figure 2.
- Some tables have insufficient supplementary text, mainly Tables 1 and 3. Noradrenaline and norepinephrine are alternative names of one substance but both names are used in Table 4. It is confusing.
- It is not clear exactly which citations are used in individual rows in Table 4. Each row should include a citation number. For example, the citation Forster et al is in the table five times and no one knows which row belongs to which of the citations 119, 117, 118, 121, and 122.
- Gh is a transglutaminase and its listing among heterotrimeric G proteins is misleading. The same applies to the use of the abbreviation hhGαh from citation 45.
- Citation 100 is mainly focused on the detection of mRNA of individual α adrenergic receptor subtypes. Binding experiments are only a minor part of the work. This publication is incorrectly cited in chapter 4.3.2. Protein expression. The combination of the words "8.7 fmol/mg protein total mRNA" does not make sense and the second half of the sentence concerns the comparison of mRNA, not protein expression.
- There are some errors in citation numbers in the paragraph between lines 478 – 499. Citations 164 and 165 do not agree with the text. Both publications deal with completely different topics.
Author Response
“I have one comment on the overall concept of the text. This manuscript includes publications since 1984. It is very commendable. On the other hand, some of the information used may already be outdated. For example, the information "In contrast, α1-AR coupling to a GTP binding protein was only detected in pathological, but not in healthy human hearts" taken from a publication from 1989 would deserve more attention. The association of α1 adrenergic receptors with G protein has been based on binding studies only.”
Any study being retrieved by our search strategy and considered applicable has been included. In some cases, such as Gh, we did not retrieve any more recent studies. However, where applicable we have added more recent data to the information from dated studies.
“The text states that it is based on 32 publications on the heart and 32 on blood vessels. In contrast, in Figure 1, the total number of used publications is 65.”
We have corrected the mistakes in the main text (page 3, lines: 107-114). The included studies were 65 in total; 33 unique cardiac reports, 31 unique vascular reports and 1 study that was reviewed for both cardiac and vascular α1-ARs.
“Figure naming should be below the Figure. Figures lack any explanatory text.”
Figure naming has been now moved below the corresponding figures. Additional explanatory text has also been added below the figures.
“PKC can be activated by calcium ions too, in Figure 2.”
Figure 2 has been corrected. The interaction between calcium ions and PKC, applying to some but not other isoforms of PKC, is now included.
“Some tables have insufficient supplementary text, mainly Tables 1 and 3.”
We are sorry for the confusion. We now avoid using abbreviations in the Table legends, and provide information on what type of tissue or vessel was used in all Tables.
“Noradrenaline and norepinephrine are alternative names of one substance but both names are used in Table 4. It is confusing.“
Both names have been removed from the table. The table has also been updated with additional text to make it more understandable.
“It is not clear exactly which citations are used in individual rows in Table 4. Each row should include a citation number. For example, the citation Forster et al is in the table five times and no one knows which row belongs to which of the citations 119, 117, 118, 121, and 122.”
We have included all corresponding references in the tables as well to provide more clarity.
“Gh is a transglutaminase and its listing among heterotrimeric G proteins is misleading. The same applies to the use of the abbreviation hhGαh from citation 45.”
Misleading statements about Gh have been corrected.
“Citation 100 is mainly focused on the detection of mRNA of individual α adrenergic receptor subtypes. Binding experiments are only a minor part of the work. This publication is incorrectly cited in chapter 4.3.2. Protein expression. The combination of the words "8.7 fmol/mg protein total mRNA" does not make sense and the second half of the sentence concerns the comparison of mRNA, not protein expression.”
The mistake in the wording of the sentence has been corrected. The missing citation referring to the protein data from the left ventricle has also been included.
“There are some errors in citation numbers in the paragraph between lines 478 – 499. Citations 164 and 165 do not agree with the text. Both publications deal with completely different topics.”
The citation error in these lines has been corrected. Citations 164 and 165 have been again checked with regards to their content we believe that they agree with the text.
Round 2
Reviewer 3 Report
Problematic parts have been fixed. Changes in the text and tables have contributed to a better understanding. I have only a few comments. Please use the same label for Transglutaminase on lines 241 and 258. The legend in Table 3 is still just "X: unchanged".
My comment on citation 47 was misunderstood. The sentence from the first version of the manuscript "In contrast, the binding of α1-AR to GTP-binding protein was detected only in pathological but not in healthy human hearts" is exactly according to publication 47. The authors of this citation deduced this conclusion from one type of experiment only. It was the effect of GTP on the competition curve. I would not have used this information without further study of the literature. For example, in 1989, individual subtypes of α1 receptors were not yet known. However, the systematic review aims to collect relevant data and this has been accomplished. It is up to the reader how he will work with the collected information.
I recommend the manuscript for acceptance.